# Development of a Tetrazolium-Derived Paper-Based Diagnostic Device as an Early, Alternative Bacteria Screening Tool

**DOI:** 10.3390/mi13010044

**Published:** 2021-12-28

**Authors:** Michael Muljadi, Chao-Min Cheng, Ching-Ju Shen

**Affiliations:** 1Institute of Biomedical Engineering, National Tsinghua University, Hsinchu 300, Taiwan; michael.muljadi@gmail.com (M.M.); chaomin@mx.nthu.edu.tw (C.-M.C.); 2Department of Obstetrics and Gynecology, Kaohsiung Medical University Hospital, Kaohsiung 807, Taiwan

**Keywords:** bacterial detection, paper-based diagnostics, MTT-PMS, *Escherichia coli*

## Abstract

(1) Background: The complexity, amount of time, and the large amount of resource required to perform gold-standard bacteria culture procedures makes it difficult to perform timely pathogenic analyses, especially in areas where such resources are not readily available. A paper-based biochemical analytical tool can potentially tackle problems economically in terms of time and convenience, potentially finding utility in applications where simple and timely detection of bacteria is necessary; (2) Methods: The utility of paper-based MTT-PMS strips was tested using a simple colorimetric analytical methodology; (3) Results: Sufficient evidence was obtained to suggest that the strips can potentially be used as a rapid and convenient early, alternative bacteria screening tool for a variety of applications; (4) Conclusions: The potential of strips for the rapid detection of bacteria compared to standard bacteria culture is a key advantage in certain clinical, agricultural, and environmental applications.

## 1. Introduction

The study and development of new technologies for the rapid screening of bacteria has been gaining traction in recent years due to their simplicity and efficiency for applications in the fields of medicine, biotechnology, and agriculture, among others. In epidemiology and public health, for example, the growing problem of antibiotic resistance can benefit from quick quantification of bacteria by helping clinicians administer suitable dosages of antibiotics to treat bacterial infections. In the case of upper urinary tract infections (UTI), it has been shown that prolonged administration of antibiotics is unnecessary for treatment [1], yet, at the same time, improper targeting could potentially cause the infection to progress further. The current gold standard for quantification and detection of bacteria is the use of traditional culture methods, which is not always possible due to the required level of training, the need for high-end equipment, and, most importantly, time constraints. For UTI, timely diagnosis and treatment has been shown to be crucial in preventing death, especially in patients with sepsis [2]. This becomes especially problematic in places such as the developing world, where resources to perform such analyses are scarce.

Rapid quantification of bacteria can also be crucial in the case of preventative healthcare and management. Taking pregnancy as an example, the presence of significant amounts of bacteria in urine (>10^5^ CFU/mL) in the absence of UTI symptoms [3], also known as asymptomatic bacteriuria (ASB), is often a precursor to UTI, which can in turn lead to adverse clinical outcomes [4,5]. In preventing the onset of UTI, the costs associated with routine screening for ASB through traditional culture procedures makes it difficult to achieve. One study revealed that routine screening for ASB was only cost-effective if rates were higher than 9% [6]. Such cost-effectiveness dependency on incidence rate makes clinical management difficult to justify, further complicating efforts in the field of disease prevention.

Apart from medicine, other industries can also benefit from rapid quantification of bacteria. A study undertaken by Cavaiuolo and colleagues in 2013, for example, developed an optimized ELISA-based biochemical assay for the detection of *Listeria monocytogenes* and *E. coli* in fresh vegetables. The limit of detection (LOD), the lowest concentration that can be measured with statistical significance, was found to be approximately 10^3^ CFU/g, with a bacteria isolation period of approximately 1–7 h. Such findings are significant in the food industry, as fresh vegetables tend to be contaminated with bacteria, and lengthy standard culture methods may render the vegetables unfit for consumption even before results are made available [7]. Another recent study by Liao and colleagues looked at the use of a tetrazolium-based MTT-PMS portable biochemical reagent and device for the detection of *E. coli* in water sources [8]. The developed portable colorimetric device was similar in rationale in that there is a need for a convenient and easy-to-use tool in areas where access to clean water is scarce. The device had an LOD of approximately 3.43 × 10^5^ CFU/mL in phosphate-buffered saline (PBS) and 1.41 × 10^6^ CFU/mL in water in less than 15 min, making it a potential avenue for the immediate elimination of unsuitable water sources. 

Similar to the Liao study [8], the use of tetrazolium dye in biochemical and other applications is due to its unique ability to reduce into formazan, a brightly colored derivative with the potential to measure cells’ metabolic activity. The term owes itself to the salt’s chemical structure, a heterocyclic compound with four nitrogen atoms in the ring. Its reduction is enzymatic, where there is cleavage of the tetrazole ring, resulting in an intensely-colored formazan [9]. MTT is a type of tetrazolium salt that was developed for use from 1957 onwards and has since been widely used in biochemical assays and other applications [10]. For example, a study in 2001 looked at the use of an MTT-based biochemical assay as an alternative method to measure fungal spore viability [11]. It was found that for a given number of viable cells, higher concentrations of MTT resulted in higher yields of formazan crystal, with the reduction based on the fungal spores’ dehydrogenase activity (DHA). 

Grela, Kozlowska, and Grabowiecka provided a fresh perspective from multiple studies on the use of MTT (3-(4,5-dimethylthiazol-2-yl)-2,5-diphenyltetrazolium bromide) tetrazolium assays as a means to assess culture viability through measurable enzymatic activity [12]. Aside from Stentelaire’s 2001 MTT-based biochemical assay to measure fungal spore viability [11], procedures involving MTT have also been used with mammalian cell lines. Mitochondrial membrane potentials, for example, have also been measured using MTT in conjunction with rhodamine B, making use of the tetrazole ring’s positive charge and differences in membrane potential [13]. However, this only further supports MTT reduction and therefore its utility in eukaryotic cells. Despite this, Grela and colleagues argued that although MTT reduction in bacteria is still poorly understood [14], it can still be utilized given proper optimization [12]. A 2014 study by Brambilla and colleagues on *Streptococcus mutans*, for example, found that their MTT assays produced accurate results that were similar to traditional plate counts following proper execution [15], in addition to being much more rapid in producing results. One of the factors affecting MTT use on prokaryotes include the incubation period, which has been shown to turn into a colorless derivative upon prolonged incubation [16]. The effectiveness of its use was demonstrated by the Liao study, successfully utilizing MTT-PMS based reagent to measure *E. coli* metabolic activity following proper incubation period optimization [8]. There is, therefore, potential for new applications in the use of MTT on prokaryotes, given properly implemented methodologies and optimizations.

*E. coli* cells can be regarded as systems of dehydrogenases, making MTT a potential tool in the measurement of their viability (Figure 1). A study by Oh and Hong in 2021 found that their MTT assay showed sufficient sensitivity to early growth phase bacterial cells compared to the plate counting method [17]. Moreover, it was found that MTT reached peak reduction within 15–30 min, and underwent no further reduction after 60 min, further supporting Liao’s procedure, and highlighting MTT’s potential for bacterial rapid screening capabilities in clinical settings. Despite its utility in rapid screening, however, biochemical assays are still problematic in terms of convenience due to the need for high-end equipment and trained personnel.

The advent of paper-based analytical devices provides many opportunities for the development of point-of-care diagnostic tools that require minimal to no resources to achieve their objectives. Highlighting the importance of early intervention for the treatment of UTI, Shih and colleagues utilized a paper-based ELISA for the rapid detection of *E. coli* [18]. Analysis was performed through colorimetric analysis using a smartphone camera. Similarly, Wang and colleagues developed a turntable paper-based ELISA to produce results within 5 h [19], incorporating Whatman Fusion Paper 5^TM^ in its design. Both studies capitalized on biochemical assays’ rapid screening potential, along with paper-based diagnostic devices’ convenience, with significant reductions in cost.

A study by Tsao and colleagues in 2020 looked at the feasibility of paper based MTT-PMS strips for point-of-care semen analysis [20], combining the MTT’s propensity for colorimetric analysis, and paper-based analytical devices’ ease of use and convenience. Using MTT-PMS embedded in a test pad on paper ribbon, Tsao and colleagues utilized the tetrazolium’s reductive property to infer sperm motility. Their rationale was to develop a convenient device that can be used as a cheap, routine screening tool for private, in-home assessment through visual cues. 

This study therefore sought to assess the feasibility of paper-based MTT-PMS strips in the rapid screening of bacterial concentration through colorimetric analysis for potential clinical, agricultural, and the environmental applications, among others, in the future. This builds on the already established potential use of MTT strips in point-of-care semen analysis.

## 2. Methodology

### 2.1. Sample Preparation

The bacterium *Escherichia coli* (*E. coli*) was used to test the feasibility of the MTT-PMS test strips. The bacteria were cultivated in Tryptic Soy Broth (TSB) in a 38.5 °C incubator and aerated under 200 rpm of shaking to ensure sufficient nutrient circulation until they were ready to use. Optical density (OD) spectrophotometry was used to estimate the concentration of bacteria measured under a wavelength of 600 nm. Four samples of bacteria suspension were prepared with distinct concentrations: 10^8^, 10^6^, 10^4^, and 10^2^ colony forming units (CFU)/mL. A control sample was also prepared using a sterilized phosphate-buffered saline (PBS) in exchange for bacteria suspension.

### 2.2. Test Strips

The test strip is an 8 × 0.5 cm paper ribbon with a 5 × 10 mm MTT-PMS-coated paper strip attached to one of the ends (Figure 2). Whatman Fusion 5^TM^ paper was used as the test pad due to its satisfactory absorption capabilities, wicking time, area, and homogeneous color development. This was based on a 2015 study by Shih and colleagues, using the paper for their paper-based ELISA following tests with 10^9^ cells/mL concentration of *E. coli* DH-5α [18]. MTT was purchased from Sigma-Aldrich (No.: M2128, USA). Similar to Tsao’s study, MTT test strips were stabilized using 0.75 mM MTT and 5% sucrose on the test pads [20,21]. The fabrication process was based on a 2020 study by Sung and colleagues [22]. The test pads were first attached to the ribbon strips. They were then submerged into the diagnostic reagent for approximately 5 min. Following submersion, test strips were then dried in a drying room under room temperature with no light for 24 h, due to the PMS’s sensitivity to light (Figure 3). They were then stored in small, non-transparent, moisture-free pouches to maintain their quality, where they can be used for up to approximately one year following fabrication [20].

### 2.3. Measurements and Data Processing

Colorimetric data was recorded using a smartphone camera under bright white light. A commercially available smartphone application, ColorPicker, was used to obtain RGB data on a Samsung Galaxy Note 9 smartphone. A rectangular region of interest (ROI) was used to detect RGB data from the MTT-PMS cotton strips (***R*_1_**, ***G*_1_**, ***B*_1_**) following submersion and incubation. RGB data of the white ribbon strip (***R*_0_**, ***G*_0_**, ***B*_0_**) (Figure 4) were also collected for standardization of results. RGB data of each strip were then converted into standardized greyscale to reflect their light intensity using the following formula:$$Y^{*}=0.299R^{*}+0.587G^{*}+0.114B^{*}$$
where:$$\begin{array}{c} R^{*}=\left|\left(R_{1}-R_{0}\right)\right|\\ G^{*}=\left|\left(G_{1}-G_{0}\right)\right|\\ B^{*}=\left|\left(B_{1}-B_{0}\right)\right|\; \end{array}$$

Repeated testing (*n*) was performed for each concentration of bacteria suspension (*c*), 10^8^, 10^6^, 10^4^, 10^2^, and the control sample, using the MTT-PMS test strips. For each suspension concentration, average standardized greyscale values were calculated using standardized greyscale values obtained through colorimetric observation:$$\bar{Y_{c}^{*}}=\frac{\sum_{i=1}^{n}Y_{i}^{*}}{n}\;$$
where $\bar{Y_{c}^{*}}$ is the average standardized greyscale value, $\bar{Y^{*}}$ for each of the different suspension concentrations. They were then plotted against the log of concentration to obtain an estimated standard curve using a linear line of best fit.

Limit of detection (LOD) of the paper-based MTT device was calculated using the following formula:$$LOD=\bar{Y_{0}^{*}}+3\sigma_{0}\;$$
where $Y_{0}^{*}$ and $\sigma_{0}$ are the average and standard deviation of the standardized greyscale values of the control samples (PBS), respectively.

### 2.4. Procedure

Bacteria suspensions of concentrations, 10^8^, 10^6^, 10^4^, 10^2^, and sterilized PBS were first prepared in sample tubes and mixed with 0.1 M Tris-EDTA solution at a 1:1 ratio to facilitate the breakdown of cell walls [23]. The solutions were incubated for 5 min. Following incubation, MTT-PMS test strips were submerged for 2 min. Strips were then taken out and let dry for 20 min away from direct light due to the MTT’s sensitivity to light [24]. Following this 20-min incubation, the MTT-PMS strips were taken out of the tube and their colorimetric data recorded using a conventional smartphone application, ColorPicker. A quick schematic of the procedure is shown in Figure 5.

## 3. Results

### 3.1. Incubation Period

A 20-min incubation was selected based on early tests made under different time periods: 0, 10, 20, and 30 min. A time of 20 min showed the best linear fit, on average, at a sample size of *n* = 5 with an *R*^2^ value of 98.48% compared to 67.25% at 10 min and 85.99% at 30 min (Figure 6).

### 3.2. Error Margins

Quick and simple error margin analyses were performed for the period of incubation and colorimetric analysis using different smartphones to examine the device’s robustness under small deviations in the standard procedure. Two additional datasets were obtained under a 5-min deviation from the standard 20-min procedure: 15 min and 25 min (Figure 7).

Additionally, two-tailed *t*-tests of two samples assuming unequal variances were performed to examine differences between the 15- and 20-min incubation periods, and the 20- and 25-min incubation periods. The *t*-tests comparing average standardized greyscale values were performed for all datasets, control, 10^2^, 10^4^, 10^6^, and 10^8^ CFU/mL, for each period of incubation (Table 1).

Based on this quick observation, there was insufficient statistical evidence to suggest a difference in average standardized greyscale values at 20 and 20 ± 5 min of incubation. This suggests that on average, observations within ±5 min of the standard incubation period may still fall within the error margins of the calibration curve.

Moreover, two-sample *t*-tests assuming unequal variances were performed on standardized greyscale data for 5 samples (*n* = 5) of 10^8^ CFU/mL *E. coli* suspensions between two different smartphones: Samsung Galaxy Note 9 and iPhone 13 (Figure 8). Two-tailed *t*-statistic of the test was found to be *t* = −2.16, with a critical value of *t* = 2.30, and a *p*-value of *p* = 0.062 (*p* > 0.05), suggesting that, on average, there was insufficient evidence to suggest that standardized greyscale values differed across different devices based on an analysis of two different smartphone camera lenses.

### 3.3. Calibration Curve

A linear estimation of the mean standardized greyscale values predicted the following line of best fit:$$\bar{Y^{*}}=0.5907x^{*}+151.81$$
where $x^{*}$ is the estimated log of bacteria concentration for each estimated standardized greyscale value $Y^{*}$. The average standardized greyscale value of the PBS control solution (μ = 151.58, σ = 0.85) was used to infer the limit of detection (LOD) of the MTT-PMS test-strip. LOD was found to be 2.05 × 10^4^ CFU/mL through calculations using a multiple of three standard deviations (3σ = 2.55). The calibration curve is shown in Figure 9.

## 4. Discussion

Based on simple colorimetric detection and analysis, there was sufficient evidence to suggest that there is potential for the MTT-PMS strips to be used in distinguishing concentrations of bacteria in a sample. Linear modeling of the relationship between the average standardized greyscale value against the log of bacteria concentration showed a statistical measure of fit (*R*^2^) of approximately 98%. This suggests that there was enough evidence to suggest that variations in the average standardized greyscale value can be sufficiently explained by differences in the log of bacteria concentration. Moreover, an F-test of the coefficients of the model was conducted against a null of no model predictive capability and was found to be statistically significant at *p* < 0.001. This suggests that there was sufficient statistical evidence to suggest a relationship between average standardized grayscale values and the log of bacteria concentration, further supporting the viability of the MTT-PMS strips and their simplistic mode of detection.

Early tests also revealed an optimal incubation period of 20 min (Figure 6). This is somewhat consistent with a 1995 study by Stowe and colleagues, where it was found that prolonged formazan incubation can result in transformation to a colorless derivative [16], potentially causing a misinterpretation of results in a colorimetric analysis. This is further supported by the findings of Wang and colleagues in the development of an improved MTT reduction assay for the evaluation of viable *E. coli* cells, where rapid formazan crystal formation was found within the first 20 min of incubation, after which the speed of its formation began to deteriorate [25]. This makes it an optimal period of incubation and hence why it was chosen for the standard procedure.

Limits of detection (LODs) of the MTT-PMS strips were also inferred based on average standardized greyscale value of the strips in sterilized PBS. Calculations based on a multiple of three standard deviations showed that LOD was approximately 2.05 × 10^4^ CFU/mL, suggesting that there was sufficient evidence to potentially screen bacteria concentrations of above 2.05 × 10^4^ CFU/mL. Although this is not to be confused with accurate quantification, the sufficiently accurate screening capability of such a device can potentially be used in applications where rapid diagnosis is crucial. In the case of asymptomatic bacteriuria, for example, the presence of more than 1 × 10^5^ CFU/mL of bacteria in the absence of symptoms of upper urinary tract infections can be classified as such. In the case of water and food safety, in Egypt, for example, it is known that the standard acceptable bacterial count for fresh chilled fish is approximately 10^6^ CFU/mL [26]. At the current device’s LOD, there is also potential for its use in the preliminary elimination of unacceptable produce prior to consumption.

The use of the MTT-PMS strips for this purpose has significant implications. Clinically, one of them being its potential utility in preventing the development of pyelonephritis during pregnancy. As was seen in a previous 1967 study, early clinical management of asymptomatic bacteriuria was shown to be an effective strategy in the prevention of pyelonephritis [27], which has been found to be detrimental during pregnancy [5]. However, due to the significant amount of time and skill required for testing, it was also found to be non-cost-effective, further complicating efforts in its prevention [6]. Since each MTT-PMS strip costs less than 1 USD to make, it has the potential to significantly cut down on costs. Moreover, it also has the potential to ensure that, in combination with clinician’s expertise, only cases that warrant further testing can be referred to more sophisticated and detailed laboratory testing, potentially allowing more scope for early detection, treatment, and prevention of UTI’s. This is a significant improvement to gold-standard bacteria screening and detection procedures, where culturing may require up to 72 h in a laboratory with highly skilled personnel.

Similarly, in terms of water and food safety, there is an industrial need for improvements in rapid detection methods to guarantee the safety of produce for consumers, especially in ready-to-consume cut vegetables, such as leafy greens [28]. Standard culture methods that aim to detect the contamination of fresh vegetables can render them unfit for consumption even before results are made available. The study by Cavaiuolo and colleagues had a sensitivity of approximately 10^3^ CFU/g, but still at a significant cost in time and required a significant amount of laboratory resources and trained personnel to perform [7]. The use of paper-based MTT-PMS has the potential to cut down on time, with further advantages in terms of convenience and ease of use, albeit with some sacrifice in sensitivity. Nonetheless, its utilization can preliminarily eliminate produce that is potentially unfit for consumption. A quick summary of relevant biochemical analytical tools is listed in Table 2.

Comparing the MTT-PMS strips with other modes of pathogenic detection, the MTT-PMS strip fared quite well, especially considering the amount of time required to obtain results. At an LOD of approximately 10^4^–10^5^ CFU/mL, the strips performed comparatively well compared to Liao and colleagues’ MTT-based portable reagent at 10^5^–10^6^ CFU/mL. This suggests that, other than the portability of both tetrazolium-based analytical detection tools, both devices are able to screen for similar concentrations of bacteria [8]. Moreover, although the MTT-PMS strips were found to be limited in terms of LOD, compared to Cavaiuolo and colleagues’ indirect ELISA [7], the short time it takes to perform a test is comparable to the advantages offered by Shih and colleagues’ paper-based ELISA [18], where its paper-based mode of detection is more similar in technique to the current strips. Similar to Liao and colleagues, the current MTT-PMS strips only require a 20-min period of incubation to produce reliable results (Figure 6). In fact, prolonged incubation may even result in the formation of a colorless derivative, making colorimetric reading inaccurate [16]. This further highlights the strips’ advantage in terms of accessibility, convenience, and speed.

It is important to note that the method is still not superior to standard culture methods. The utility of the strips is limited to quantitative screening and does not differentiate between different pathogens, limiting its use to applications where screening is paramount. Moreover, as in the study by Tsao and colleagues, although the strips may not be able to accurately quantify bacteria concentration, they have the potential to estimate an average range based on colorimetric data. Therefore, given their ease of use, as well as their ability to operate with minimal sophisticated equipment, this study has shown that the strips can potentially be utilized as an early, alternative screening tool in clinical, environmental, and agricultural applications, among others. The results of this study has created openings for further studies to investigate the utility of paper-based tetrazolium derivatives in the quantification of other pathogens, for applications in food safety, such as *Listeria monocytogenes* as investigated by Cavaiuolo and colleagues’ [7], for water safety applications, such as Liao and colleagues [8], or even in general clinical samples, such as urine and other bodily fluids in comparison with standard culture procedures. Whatever the application, convenient and rapid screening of bacteria concentration will not only be beneficial in clinical care, but also in other fields such as agriculture and biotechnology.

## Figures and Tables

**Figure 1 micromachines-13-00044-f001:**
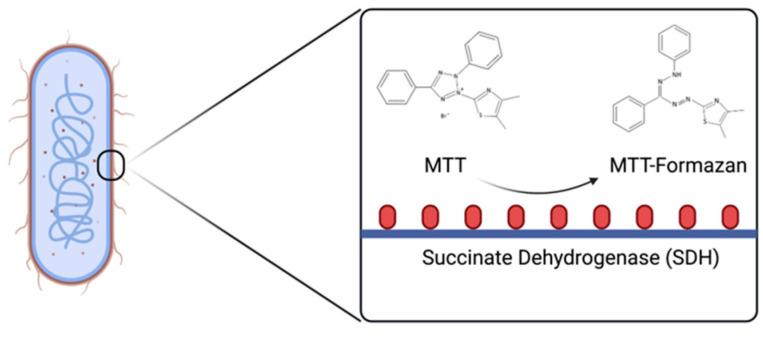
Basic mechanism of the reduction of MTT to MTT-Formazan on bacteria. Created with BioRender.com (https://app.biorender.com/). Last accessed on 9 October 2021.

**Figure 2 micromachines-13-00044-f002:**
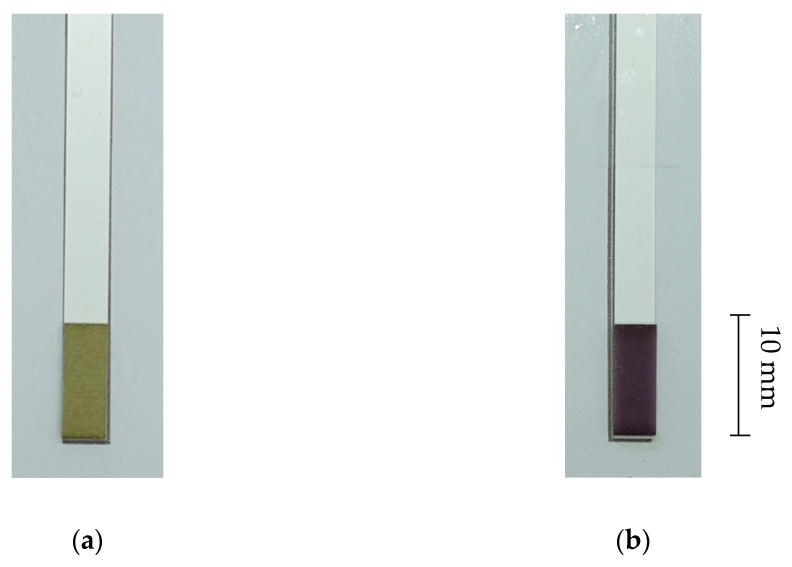
The MTT-PMS test strips: (**a**) MTT-PMS test strips prior to submersion into solution; (**b**) MTT-PMS test strips following submersion and incubation.

**Figure 3 micromachines-13-00044-f003:**
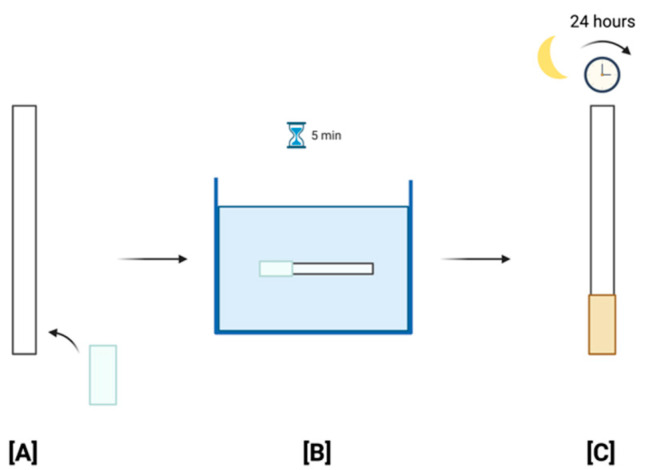
MTT-PMS test strip fabrication procedure: (**A**) Attachment of Whatman Fusion 5^TM^ paper test pad on ribbon strip; (**B**) Submersion of test strip in reagent for approximately 5 min; (**C**) Drying of the test strips in a dark room at room temperature for 24 h. Created with BioRender.com (https://app.biorender.com/). Last accessed on 6 December 2021.

**Figure 4 micromachines-13-00044-f004:**
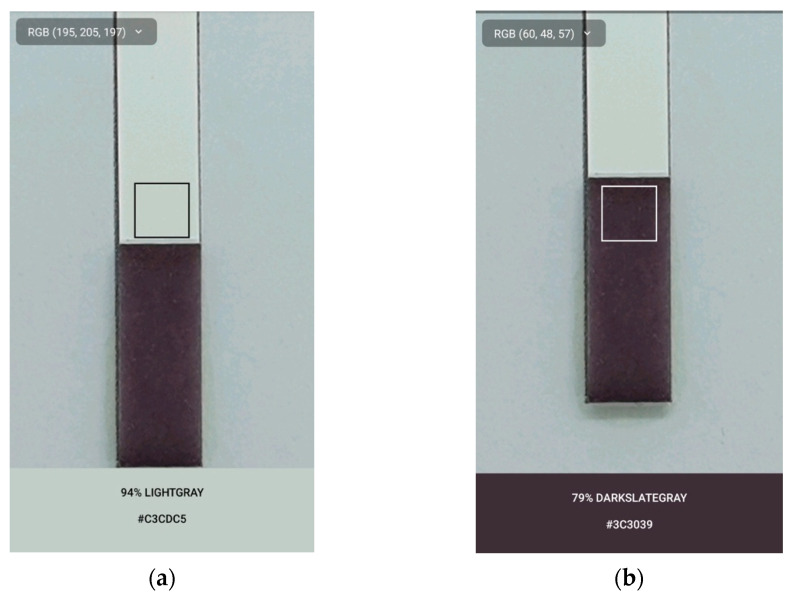
Colorimetric detection of the MTT-PMS strips following submersion into solution using the smartphone application ColorPicker: (**a**) RGB data collection of the white ribbon strips for standardization (***R*_0_**, ***G*_0_**, ***B*_0_**); (**b**) RGB data collection of the MTT cotton strips (***R*_1_**, ***G*_1_**, ***B*_1_**).

**Figure 5 micromachines-13-00044-f005:**
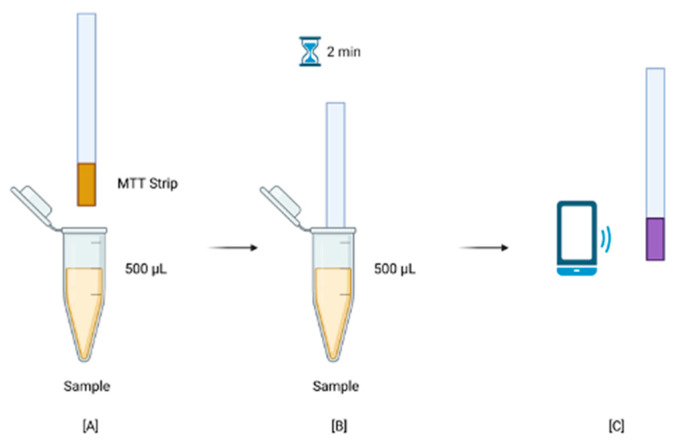
MTT-PMS test strip data collection procedure: (**A**) Preparation of bacteria suspension in a sample tube; (**B**) Submersion of MTT test strips for 2 min; (**C**) Recording of RGB colorimetric data using a smartphone camera. Created with BioRender.com (https://app.biorender.com/). Last accessed on 10 November 2021.

**Figure 6 micromachines-13-00044-f006:**
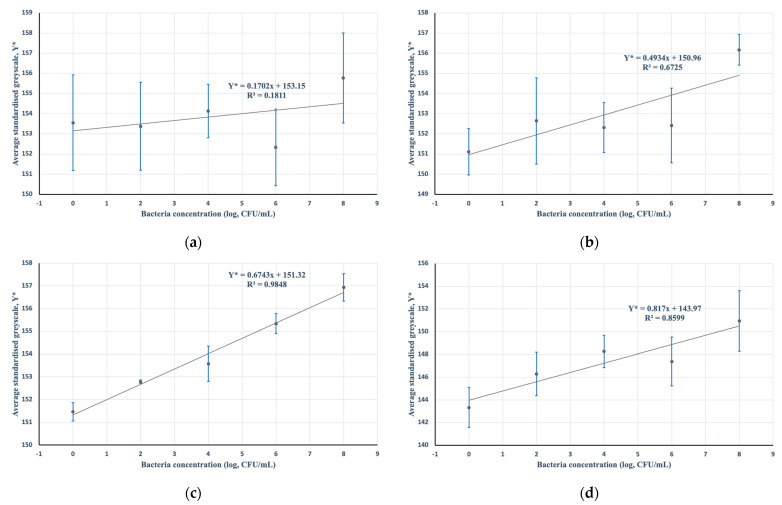
Data of the average standardized greyscale values of MTT-PMS strips against the log of bacteria concentration for different incubation periods (*n* = 5): (**a**) 0 min; (**b**) 10 min; (**c**) 20 min; (**d**) 30 min.

**Figure 7 micromachines-13-00044-f007:**
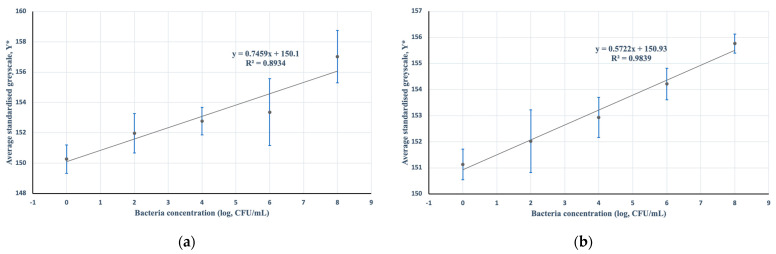
Data of the average standardized greyscale values of MTT-PMS strips against the log of bacteria concentration for two additional incubation periods (*n* = 5): (**a**) 15 min; (**b**) 25 min. F-statistic of both regressions against a null of no model predictive capability were found to be F = 25.13 (*p* < 0.05) and 183.14 (*p* < 0.001), respectively.

**Figure 8 micromachines-13-00044-f008:**
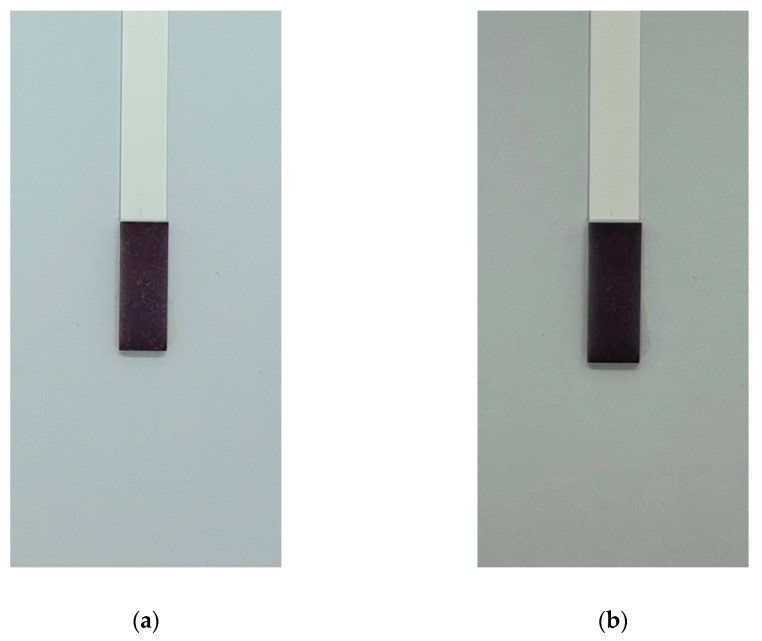
The MTT-PMS test strips for 10^8^ CFU/mL *E. coli* following a 20-min incubation under: (**a**) Samsung Galaxy Note 9; (**b**) iPhone 13.

**Figure 9 micromachines-13-00044-f009:**
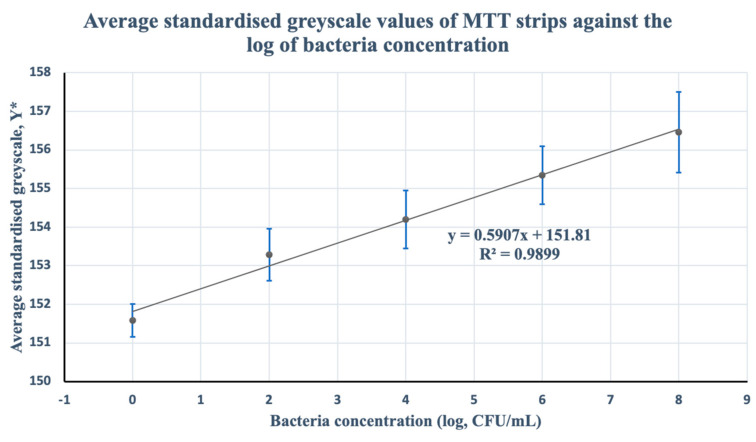
Mean standardized greyscale value of the MTT-PMS strips (*n* = 23) in different concentrations of bacteria where the vertical axis represents the average standardized greyscale value from colorimetric analysis, and the horizontal axis represents the log of bacteria concentration. F-statistic of the regression against a null of no model predictive capability was found to be F = 293.85 (*p* < 0.001).

**Table 1 micromachines-13-00044-t001:** Results summary of two-tailed two-sample *t*-tests assuming unequal variances between datasets of 15- and 20-min, and 20- and 25-min incubation periods. *p*-values of less than 0.05 were considered statistically significant.

Concentration (CFU/mL)	Incubation Period Comparison (*t*-Statistic)	
	15 vs. 20 min	20 vs. 25 min
control	1.41 (*p* > 0.05)	0.69 (*p* > 0.05)
10^2^	1.18 (*p* > 0.05)	1.21 (*p* > 0.05)
10^4^	1.67 (*p* > 0.05)	1.71 (*p* > 0.05)
10^6^	1.04 (*p* > 0.05)	1.91 (*p* > 0.05)
10^8^	−0.31 (*p* > 0.05)	1.37 (*p* > 0.05)

**Table 2 micromachines-13-00044-t002:** A short summary of studies utilizing biochemical analytical tools for the detection of pathogens.

Study	Type	Material	Target of Detection	Limit of Detection
Cavaiuolo, et al., 2013	Indirect ELISA	Reagent: - Horseradish peroxidase (HRP) - TMB substrate	- *Listeria monocytogenes* - *Escherichia coli*	10^3^ CFU/g
Liao, et al., 2020	MTT-based reagent	Reagent: - Tetrazolium salt (MTT) - Phenazine methosulfate (PMS)	*Escherichia coli*	10^5^–10^6^ CFU/mL
Shih, et al., 2015	Paper-based ELISA	Paper material: - Whatman Fusion 5TM paper	*Escherichia coli*	10^5^ CFU/mL
Stentelaire, et al., 2001	MTT-based reagent	Reagent: - Tetrazolium salt (MTT)	*Penicillium digitatum*	4 × 10^7^ spores/mL

## Data Availability

Supporting data and results are available upon request. Please contact the corresponding author(s) directly.

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
