# Peer review of "Development of a Tetrazolium-Derived Paper-Based Diagnostic Device as an Early, Alternative Bacteria Screening Tool"

_micromachines, 2021, doi:10.3390/mi13010044_

Round 1
Reviewer 1 Report
Comments to the Author:
The manuscript reports the feasibility of paper-based MTT-PMS strips for rapid screening of bacterial concentration via colorimetric analysis. The authors claim that the such method has the potential to serve as a cost-effective alternative screening tool for asymptomatic bacteriuria during pregnancy. While the concept and purpose are worth noting, the manuscript does not cover enough discussion and needs further in-depth research and data in order to be more significant. For example, since this is supposed to be a rapid test tool compared to accurate lab testing environment, what are the error margins of the conditions needed to get relative accurate readings (eg. incubation temperature and time, different camera and lens, etc)? How does this method work on actual bacteriuria samples besides prepared bacteria suspension? Also, since the LOD is at 2.05 x 104 CFU/ml, the number of meaningful data points seems limited.
Reviewer 2 Report
In the current study, the authors demonstrate the utility of tetrazolium-derived paper strips (MTT-PMS strips) in the rapid screening of bacterial concentration. The MTT (3-(4,5-dimethylthiazol-2-yl)-2,5-diphenyltetrazolium bromide) tetrazolium assay is a common tool in assessing the metabolic activity of the living cells. The enzymatic reduction of the lightly colored tetrazolium salt results in the formation of formazan, which has an intense purple-blue color that can be quantified in diverse ways.
The article is well structured into section and subsections and professionally written. It is within the scope of journal and will be of interest to the readers.
However, there are some concerns regarding the literature covered in the introduction and the discussion section. There are interesting studies available that discuss the utilities of MTT-PMS based diagnostics or MTT assays in general. It will be appropriate to incorporate them. For instance, the following articles:
Moussa, S. H., Tayel, A. A., Al-Hassan, A. A. & Farouk, A. Tetrazolium/Formazan Test as an Efficient Method to Determine Fungal Chitosan Antimicrobial Activity. Journal of Mycology 2013, 1–7 (2013).
Hundie GB, Woldemeskel D, Gessesse A. Evaluation of Direct Colorimetric MTT Assay for Rapid Detection of Rifampicin and Isoniazid Resistance in Mycobacterium tuberculosis. PLoS One. 2016 Dec 28;11(12):e0169188. doi: 10.1371/journal.pone.0169188. Erratum in: PLoS One. 2017 Feb 6;12 (2):e0171964. PMID: 28030634; PMCID: PMC5193450.
Singh AT, Lantigua D, Meka A, Taing S, Pandher M, Camci-Unal G. Paper-Based Sensors: Emerging Themes and Applications. Sensors (Basel). 2018 Aug 28;18(9):2838. doi: 10.3390/s18092838. PMID: 30154323; PMCID: PMC6164297.
Hosseini S, Vázquez-Villegas P, Martínez-Chapa SO. Paper and Fiber-Based Bio-Diagnostic Platforms: Current Challenges and Future Needs. Applied Sciences. 2017; 7(8):863. https://doi.org/10.3390/app7080863
Tsao YT, Yang CY, Wen YC, Chang TC, Matsuura K, Chen Y, Cheng CM. Point-of-care semen analysis of patients with infertility via smartphone and colorimetric paper-based diagnostic device. Bioeng Transl Med. 2020 Aug 18;6(1):e10176. doi: 10.1002/btm2.10176. PMID: 33532582; PMCID: PMC7823130.
Grela E, Kozłowska J, Grabowiecka A. Current methodology of MTT assay in bacteria - A review. Acta Histochem. 2018 May;120(4):303-311. doi: 10.1016/j.acthis.2018.03.007. Epub 2018 Mar 30. PMID: 29606555.
Stockert JC, Horobin RW, Colombo LL, Blázquez-Castro A. Tetrazolium salts and formazan products in Cell Biology: Viability assessment, fluorescence imaging, and labeling perspectives. Acta Histochem. 2018 Apr;120(3):159-167. doi: 10.1016/j.acthis.2018.02.005. Epub 2018 Feb 26. PMID: 29496266.
Buranaamnuay K. The MTT assay application to measure the viability of spermatozoa: A variety of the assay protocols. Open Vet J. 2021 Apr-Jun;11(2):251-269. doi: 10.5455/OVJ.2021.v11.i2.9. Epub 2021 May 8. PMID: 34307082; PMCID: PMC8288735.
Dou M, Sanjay ST, Benhabib M, Xu F, Li X. Low-cost bioanalysis on paper-based and its hybrid microfluidic platforms. Talanta. 2015 Dec 1;145:43-54. doi: 10.1016/j.talanta.2015.04.068. Epub 2015 May 6. PMID: 26459442; PMCID: PMC4607929.
Page 1, Line 26, 37: Escherichia coli (E. coli) needs to be italicized.
Similarly in page 2, Line 52; page 3, Line 96; page6, Line 161: E. coli needs to be italicized.
Reviewer 3 Report
The manuscript entitled "Development of a tetrazolium-derived paper-based diagnostic device as an alternative bacteria screening tool" reports a new test-strip methodology for detecting bacteria in urine samples.
The manuscript is well-written and justifies the use of the test strips coated with MTT. Nevertheless, the presentation of the manuscript left me overall confused. I want to raise awareness of some issues that need to be resolved prior to make the manuscript considered for publication.
Main Point 1
Generally, when a manuscript presents a new methodology for detection it is necessary to introduce a proof of concept. It is well-known fact that MTT can be introduced in multi-well plates, strips, or on plates. Even more MTTs assays for bacterial screening in clinical cases (and for MDR strains) is being proposed for over 20 years now (you can check Abate et al., 1998).
The manuscript basically describes what we already know about mtt and the innovation is supposed to be the coating of MTT in stripes. How is that done?
Still, the authors use single cultures of E. coli and not urine samples.
Additionally, to my knowledge urine samples already harbor a heavy bacterial load of at least 103 and this method cannot distinguish between different bacterial strains. This is a limitation that is not addressed in the present form of the manuscript. It is also beyond the detection of the device. It is well known that mtt assays can detect both gram-negative and gram-positive bacteria.
Also, urine samples are known to have many metabolites that could interfere with MTT assay and overestimate or underestimate the bacterial load.
It is also important in a bio detector to know for how long and exactly how the strips could be preserved.
Another thing is that usually in mtt assays we use DMSO to better dissolve the formate crystals. Here authors rely on the strip without knowing how well the crystals were dissolved. This could interfere with the detection limit.
There are some more major points as well which I do believe that authors need to do a proof of concept study for start!
Some minor points to improve the manuscript:
Lines 148-154 this is not materials and methods this is discussion.
Escherichia coli need to be in italics throughout the manuscript.
Line 250 this is not a proof of concept study. This is misleading. There are no clinical samples!!
There are many experiments missing before considering this manuscript suitable for publication.
Round 2
Reviewer 1 Report
Comments to the Author: The authors have responded the queries and addressed the comments from the last version. As a result, I recommend the manuscript be accepted for publication.
Reviewer 3 Report
The resubmitted manuscript has substantial changes. The quality of the manuscript has been increased along with discussing limitations and optimization of the method.
I still do believe that a proof of concept is required in order for the manuscript to be considered for publication. Additionally, it is now evident that it is a time-consuming method as seen in figure 3, which shows that requires more than 24 hours for the end-user to have a result. Also, it seems that the end-user requires expensive and special sterile equipment to dry the paper for 24 h, reducing the initial novelty that the authors suggested! Please note all this in the discussion section.
Line 371 Authors should stop thinking that it is an alternative detection method!! It is an early detection method, complimentary of the standard protocols which give accurate and clinically important results!
Unfortunately, I have to reiterate some of my concerns!
a) Urine samples are known to have many metabolites that could interfere with MTT assay and overestimate or underestimate the bacterial load. ESPECIALLY DEAD-CELL METABOLITES which can interfere substantially in the formation of formazan crystals (both negatively and positively)
b) Another thing is that usually in mtt assays we use DMSO to better dissolve the formate crystals. Here authors rely on the strip without knowing how well the crystals were dissolved. A small discussion should be written regarding the changes between in vitro and clinical application.
I believe that all these concerns could be omitted if a proof of concept was introduced.